# Room-temperature multiferroicity in sliding van der Waals semiconductors with sub-0.3 V switching

Rui Chen[1,2,12], Fanhao Meng[1,2,12] ✉, Hongrui Zhang[1,3,12], Yuzi Liu[4,12], Shancheng Yan[5,12], Xilong Xu[6], Linghan Zhu[6], Jiazhen Chen[1,2], Tao Zhou[4], Jingcheng Zhou[1], Fuyi Yang[1,2], Penghong Ci[1,2], Xiaoxi Huang[1], Xianzhe Chen[1,2], Tiancheng Zhang[1], Yuhang Cai[1,2], Kaichen Dong[1,2], Yin Liu[7], Kenji Watanabe[8], Takashi Taniguchi[8], Chia-Ching Lin[9], Ashish Verma Penumatcha[9], Ian Young[9], Emory Chan[10], Junqiao Wu[1,2], Li Yang[6], Ramamoorthy Ramesh[1,2,11] & Jie Yao[1,2] ✉

The search for van der Waals (vdW) multiferroic materials has been challenging but also holds great potential for the next-generation multifunctional nanoelectronics. The group-IV monochalcogenide, with an anisotropic puckered structure and an intrinsic in-plane polarization at room temperature, manifests itself as a promising candidate with coupled ferroelectric and ferroelastic order as the basis for multiferroic behavior. Unlike the intrinsic centrosymmetric AB stacking, we demonstrate a multiferroic phase of tin selenide (SnSe), where the inversion symmetry breaking is maintained in AA-stacked multilayers over a wide range of thicknesses. We observe that an interlayer-sliding-induced out-of-plane (OOP) ferroelectric polarization couples with the in-plane (IP) one, making it possible to control out-of-plane polarization via in-plane electric field and vice versa. Notably, thickness scaling yields a sub-0.3 V ferroelectric switching, which promises future low-power-consumption applications. Furthermore, coexisting armchair- and zigzag-like structural domains are imaged under electron microscopy, providing experimental evidence for the degenerate ferroelastic ground states theoretically predicted. Non-centrosymmetric SnSe, as the first layered multiferroic at room temperature, provides a novel platform not only to explore the interactions between elementary excitations with controlled symmetries, but also to efficiently tune the device performance via external electric and mechanical stress.

The emerging ferroic orders in 2D vdW materials have spurred intensive research interest from both fundamental physics and device application perspectives[1–10]. Multiferroic materials, possessing multiple collective state switches such as ferroelectricity, ferromagnetism, ferroelasticity and ferrotoroidicity, allow for the implementation of correlated-electron systems into state-of-the-art devices, showing tremendous potential for attojoule logic computing and memory[11–14]. In the community of multiferroics, 2D vdW materials have gained particular attention owing to their ideal interfaces, fundamentally new physics, miniaturized device footprint, flexibility of fabrication, to

name a few. However, establishing two or more ferroic order parameters especially at the 2D limit is faced with serious challenges, and so far experimental reports have been scarce. Recently, monolayer $NiI_2$ was experimentally demonstrated to be a type-II magnetoelectric multiferroic system at cryogenic temperatures, bridging the gap between multiferroics and 2D materials[15]. In parallel to that, several other 2D multiferroics are theoretically proposed[16–20], which highlight not only magnetoelectric coupling, but also potential association with ferroelasticity.

Layered group-IV monochalcogenides (MXs, M: Ge, Sn; X: S, Se) are orthorhombically distorted from the cubic structure, which yields extraordinary crystal anisotropy similar to phosphorene, and thus fascinating unconventional optical and thermoelectric properties[21–27]. MXs have been proposed to be a promising multiferroic system, potentially hosting both ferroelectricity in the monolayer and ferroelasticity in the bulk[19,20]. There are four energetically degenerate multiferroic ground states, which results from the spontaneous tensile strain along $x$ and $y$ relative to the cubic paraelectric (paraelastic) structure. Figure 1d depicts the free energy landscape as a function of lattice distortion $\varepsilon_x$ ($\varepsilon_y$). The electric field $E_{\pm x}$ ($E_{\pm y}$) drives the ferroelectric switching along $x$ ($y$), whereas the mechanical stress $\sigma_x$ ($\sigma_y$) leads to the ferroelastic transition into the $x$ ($y$) axis. As a consequence, charge and lattice degrees of freedom in MXs are cross-linked through the four degenerate ground states, and with the non-volatile switching by external fields, novel designs of functional devices utilizing the switching of electric polarization and lattice strain can be achieved[19,20].

Experimentally, in-plane (IP) ferroelectricity was reported in monolayer and AA-stacked few-layer MX, where imaging and manipulation of the ferroelectric domains was also demonstrated[28–33]. However, OOP electric polarization which facilitates wider device applications, remains elusive in this system. In particular, the rich variety of stacking symmetry and interlayer sliding configurations in such vdW materials potentially offers a pathway to realize additional dimensions of polarization[7–10,34–38]. In this work, we report the discovery of the non-centrosymmetric stacking and interlayer sliding which give rise to co-existing and correlated IP and OOP ferroelectric properties. Such ferroelectrically stacked vdW layers offer a nearly linear scaling behavior, i.e., as the material thickness scales down, the ferroelectric switching voltage of SnSe reaches sub-0.3 V, which offers new opportunity towards future attojoule energy-efficient applications[28–31]. In addition, we also observe two types of characteristic ferroelastic domains in such SnSe nanosheets with HRTEM from the same cross section. For the first time, we experimentally demonstrate the coexistence of ferroelectric and ferroelastic orders in a 2D vdW system. The extension of 2D multiferroic family to room temperature will bring about more in-depth explorations of intrinsic coupling between ferroic orders. In contrast with previous reports on MXs[29,30,32,33,39], our work on the AA-stacked SnSe not only relaxes the stringent thickness limitation towards ferroelectricity but also experimentally demonstrates the existence of multiferroic order. Multiferroic SnSe with OOP polarization unlocks abundant degrees of freedom to tune the material figures of merit for low-power device applications, such as non-volatile memory, logic, photodetector, actuator, and so forth[11,40–44].

## Results
### AA-stacking and broken inversion symmetry
We realize a low-temperature (250 °C) physical vapor deposition (PVD) growth of AA-stacked (more evidence in Fig. 3) SnSe single crystals on a variety of substrates, including silicon ("Methods"). This low-temperature recipe allows for high compatibility with the back-end-of-line (BEOL) of complementary metal-oxide-semiconductor (CMOS) technologies. Optical microscope images of SnSe flakes on Si substrate are presented in the Supplementary Fig. S1. The thickness of as-grown SnSe varies from 5 nm to a few microns, with slight adjustment of the

growth condition as discussed in "Methods" section. Mechanical exfoliation also assists in thinning down the nanoflakes. Figure 1a shows the crystal structure of the AA-stacking phase of SnSe, where the lateral orientation of each layer is aligned (Fig. 1b). In contrast to centrosymmetric AB-SnSe (Fig. 1c), AA-stacking preserves the $C_{2v}$ point group as of the monolayer, realizing a non-centrosymmetric phase with thickness ranging up to over 100 nm as we have tested.

In order to study the symmetry of synthesized crystals, we first conduct optical second harmonic generation (SHG) measurements ("Methods")[45–47]. An 800 nm pulsed laser is incident normal to the sample plane, and the linear polarizer and analyzer are maintained parallel to each other. Strong SHG signals at 400 nm wavelength are observed, with the second-order nature confirmed by the power-dependent measurement as plotted in Supplementary Fig. S2b[45]. The strong SHG response confirms the broken inversion symmetry of the as-grown SnSe crystals, in contrast to the conventional AB-SnSe with vanishing SHG. A pair of half-wave plates placed in the incidence and detection path respectively, are rotated together to obtain the polarization-dependent SHG. An angular pattern corresponding to the $C_{2v}$ point group was clearly revealed, as shown in Fig. 1e. To gain insights into the stacking symmetry, we calculated the second-order nonlinear susceptibility of bilayer AA-SnSe (Fig. 1f, "Methods"), with the in-plane SHG response written as: $\chi_\parallel = (2\chi_{xxy} + \chi_{yxx})\sin\theta\cos^2\theta + \chi_{yyy}\sin^3\theta$. As displayed in Fig. 1e, the experimental data agree well with the calculation results, suggesting the AA stacking of our synthesized SnSe crystals with a broken inversion symmetry. Such consistency was further verified by the polarization-dependent SHG pattern at 1064 nm CW laser excitation, as shown in Supplementary Fig. S2c,d.

### IP and OOP ferroelectricity with low-voltage switching
To verify the spontaneous polarization in the non-centrosymmetric SnSe, we investigated the ferroelectric behavior through the Dual AC Resonance Tracking (DART) piezoresponse force microscopy (PFM, see "Methods"). Figure 2a presents a schematic illustration of in-plane PFM measurements. A DC voltage is applied between the tip and metal electrode, so that the horizontal electric field can switch the IP electric polarization[30]. In the meantime, an AC voltage is used to detect the piezoelectric response which is associated with the electric polarization. As shown in Fig. 2b, the typical butterfly-shaped amplitude behavior and the hysteretic phase against bias curve indicate the in-plane ferroelectricity in the SnSe sample. Such behaviors have been observed in all samples we have tested, with various thicknesses. Notably, previous attempts to identify ferroelectricity in MXs were only limited in monolayer or few layers, mainly due to the loss of AA-stacking in thicker samples[29]. Our synthesized SnSe, as a new polar crystal, overcomes such thickness restriction and show consistent ferroelectric response at all thicknesses.

In the meantime, we also observe the OOP ferroelectric responses in our SnSe crystals, which was not discussed in previous theoretical proposals. Furthermore, as the vertical electrical response has an intrinsic connection with the thickness of the samples, particular attention has been paid to identifying low switching voltage of SnSe, from the perspective of low-power device applications. As shown in Fig. 2c, the vertical electric field between the tip and heavily doped silicon substrate can flip the OOP polarization in SnSe. Figure 2d displays the representative hysteresis loops of phase and butterfly-shaped amplitude as a function of bias, indicative of the OOP ferroelectricity. The thickness of this sample is 87.1 nm, and the corresponding coercive voltage is around 2.1 V.

In light of the thickness dependence of the OOP ferroelectric switching voltage, we then explore the SnSe sheets with reduced thicknesses. By increasing the carrier gas flow and decreasing both the growth temperature and growth time, we were able to suppress the layer-by-layer growth, and thus, to substantially decrease the thickness of SnSe sheets down to nanometer scale. It is worth noting that the

coercive voltage is only 0.28 V when the thickness reaches 5 nm (Fig. 2e), which is remarkably lower than that of hafnium oxides[48] and other 2D ferroelectrics[5-10]. Figure 2f summarizes the reduction of

switching voltage as the sample thickness decreases. It shows a clear trend that the coercive voltage scales almost linearly with sample thickness. It has been reported that the depolarization field in ultrathin

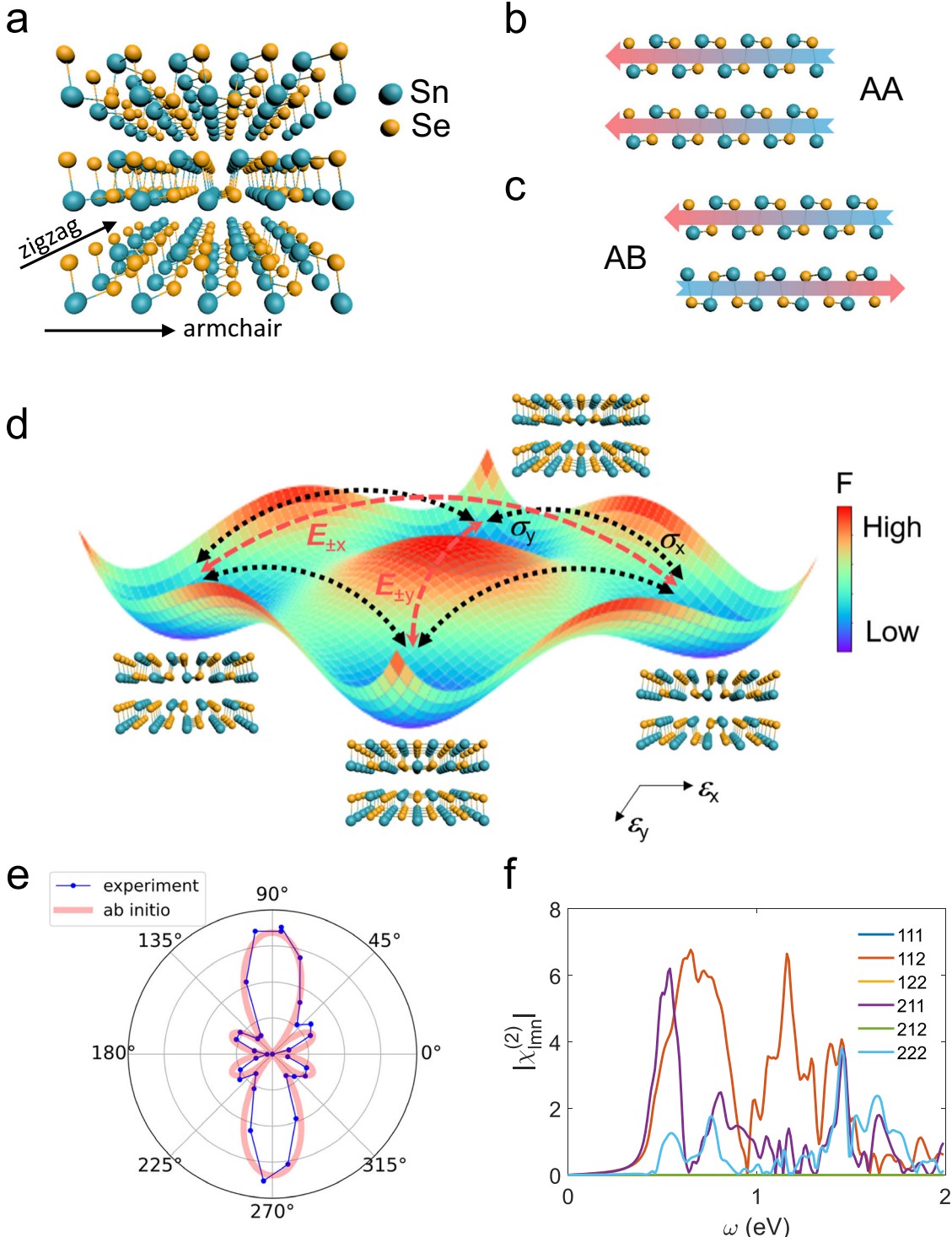

**Fig. 1 | Crystal structure and non-centrosymmetric stacking of SnSe.**
**a** Schematic crystal structure of non-centrosymmetric SnSe displaying an AA-type stacking, where every single layer is vertically aligned. Blue and yellow spheres represent Sn and Se atoms, respectively. **b**, **c** Schematic illustration of bilayer SnSe with AA and AB stacking symmetry. For centrosymmetric AB stacking, adjacent layers are antiparallel to each other. **d** Schematic of the multiferroic ground states predicted in MXs. The landscape of Landau free energy illustrates four ferroelectric (ferroelastic) states, $P_{\pm x}$ and $P_{\pm y}$, in the $xy$ plane. Non-volatile ferroelectric phase

transition along $x$ ($y$) can be triggered by the E-field $E_{\pm x}$ ($E_{\pm y}$) along $x$ ($y$). Meanwhile, the stress $\sigma_x$ ($\sigma_y$) along $x$ ($y$) drives the ferroelastic states from $P_y$ ($P_x$) into $P_x$ ($P_y$), which is also non-volatile. In that sense, electric field control of ferroelasticity and stress control of ferroelectricity are enabled in multiferroic MXs. **e** Polarization-dependent in-plane SHG of synthesized SnSe, consistent with the ab initio calculations of bilayer AA-SnSe (red curve). The polarizer and the analyzer are set to be parallel to each other. **f** DFT calculation of the in-plane second-order susceptibility response of bilayer AA-SnSe.

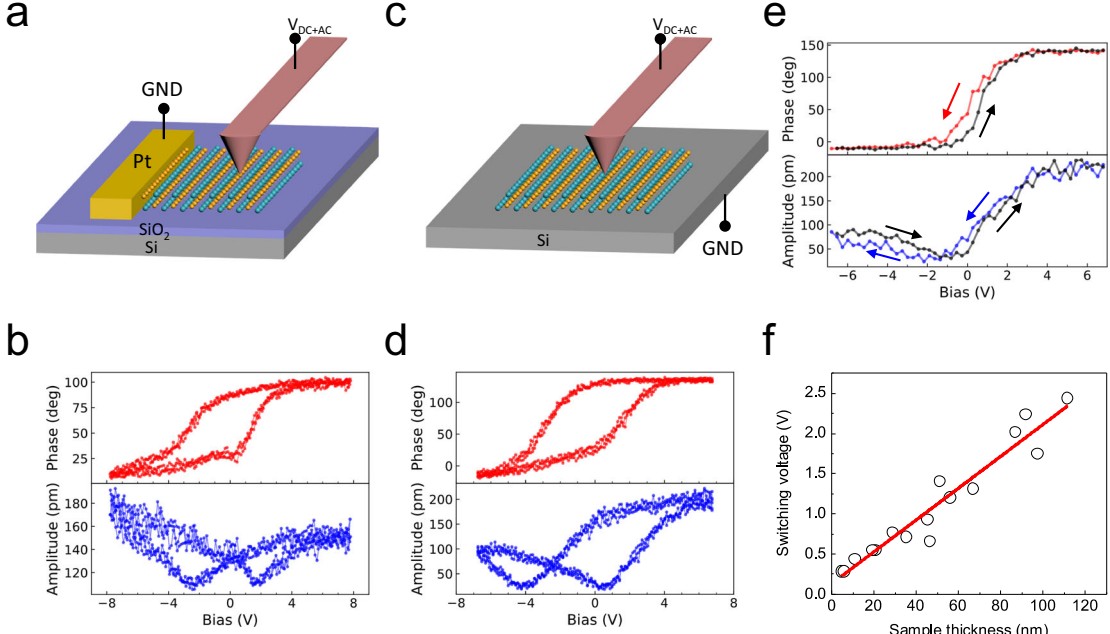

**Fig. 2 | IP and OOP ferroelectric switching verified by PFM. a** Schematic of the IP PFM measurements. SnSe nanoflakes are prepared on the 300 nm SiO₂/Si substrate. A Pt electrode is fabricated grounding one side of the SnSe sample. An IP electric field is applied between the electrode and tip for the switching and probe of local polarization. (GND: ground) **b** IP PFM phase and amplitude as a function of voltage. Non-volatile IP ferroelectricity is evidenced by the hysteresis recorded in "DC off" mode. The sample thickness is 81.6 nm. **c** Schematic of OOP PFM measurements. SnSe nanoflakes are grown on a heavily doped Si substrate, which serves as the bottom electrode. The biased tip that contacts the sample senses the OOP piezo-electric response. **d** OOP PFM phase and amplitude ("DC off" mode) as a function of voltage. Similar to the IP results in (**b**), the hysteretic switching behaviors explicitly indicate the non-volatile bulk ferroelectricity along the OOP direction. The sample thickness is 87.1 nm. **e** OOP PFM results of 5-nm-thick SnSe sheet exhibiting a low switching voltage of 0.28 V. The arrows in figure denote the sweeping directions of voltage. **f** Evolution of switching voltage as sample thickness scales down. An almost linear relationship is indicated by the red line.

ferroelectric films may suppress the coercive field[49]. This may deviate the system from the classical Janovec-Kay-Dunn (JKD) scaling, and induce the essentially thickness-independent coercive field. The linear fitting shows a slope of about 200 kV/cm, which agree with the nearly constant coercive field in samples above 35 nm (Supplementary Fig. S6). As the material further thins down, more reduction of the switching voltage is expected, although the measured coercive field tends to increase, which might pose a limit to the smallest switching voltage we can achieve. Future efforts including interface engineering, defect engineering and doping engineering could potentially lead to ultralow voltage switching in fewer layer SnSe nanosheets. Driving the lower limit of ferroelectric operation voltage will open up new avenues towards next-generation non-volatile nanoelectronic devices with ultra-low power consumption[11,41,50].

### Sliding induced OOP polarization and its coupling mechanism

Having verified the ferroelectricity in SnSe, we then analyze the underlying mechanisms through a combination of high-resolution TEM (HRTEM) and density functional theory (DFT) calculations ("Methods"). Figure 3a presents the cross-sectional TEM image with electron beam along [010] zone axis. The HRTEM image clearly shows the vertically aligned armchair-like structure and matches very well with the AA stacking model. It confirms that our as-grown SnSe flakes are AA stacked with accumulated polarization, in stark contrast with conventional MXs with centrosymmetric AB stacking and thus no spontaneous polarization[25,51]. In order to further prove the AA stacking order, we conducted selected area electron diffraction (SAED) along the [010] zone axis and compared it with simulations ("Methods"). The experimental pattern (Fig. 3b) displays the orthorhombic symmetry and high crystalline quality, and more importantly, matches well with the simulated pattern of AA-SnSe (Fig. 3c), while ruling out AB stacking (Supplementary Fig. S7). Therefore, we unambiguously unravel the AA-

stacking of SnSe with inversion symmetry breaking, which is an essential prerequisite for the existence of ferroelectricity. It should be emphasized that this is the first time an AA stacking phase stabilized in bulk MXs is observed.

As aforementioned, the superposition of monolayer polarization with AA-stacking would enable the IP ferroelectric order in multilayer SnSe. While for the OOP dipole, we show in the following that the interlayer sliding plays a dominant role, in analogy to the "sliding ferroelectricity" in other 2D materials[7,9,10,34–36,52–54]. Through analyzing the individual atomic positions and shifts in the TEM image, we are able to observe a collective interlayer sliding (Fig. 3d, e and Supplementary Fig. S9). The overall shift of Sn²⁺ and Se²⁻ ions between adjacent layers gives rise to unbalanced interlayer dipoles, resulting in a non-zero net OOP polarization (Fig. 4a). The direction of OOP polarization is determined by the sliding direction and locked to the IP polarization state, thus accounts for the observed OOP ferroelectricity. With a leftward sliding as in Fig. 4a, when the IP state is switched from +P_IP to −P_IP, the OOP dipoles flip from −P_OOP to +P_OOP correspondingly, and vice versa. Such coupling arises as a direct consequence of AA stacking and sliding structure, as the (−P_OOP, +P_IP) and (+P_OOP, −P_IP) ground states are intrinsically associated with a C_2y symmetry operation in this fashion (Supplementary Fig. S10).

To further understand the sliding induced OOP polarization and the coupled switching process, we perform first-principles DFT calculations of a bilayer AA-SnSe structure. By varying the interlayer shift while relaxing the atomic structures, we observe a double-well-shaped profile of the total energy (Fig. 4b). Two local energy minima with an opposite OOP polarization are identified aside the energy maximum (corresponding to the non-polarized intermediate state), and the system tends to relax into either of these energy-favorable states, agreeing well with the experimentally observed collective sliding structure. Following the ferroelectric switching process depicted in Fig. 4a with

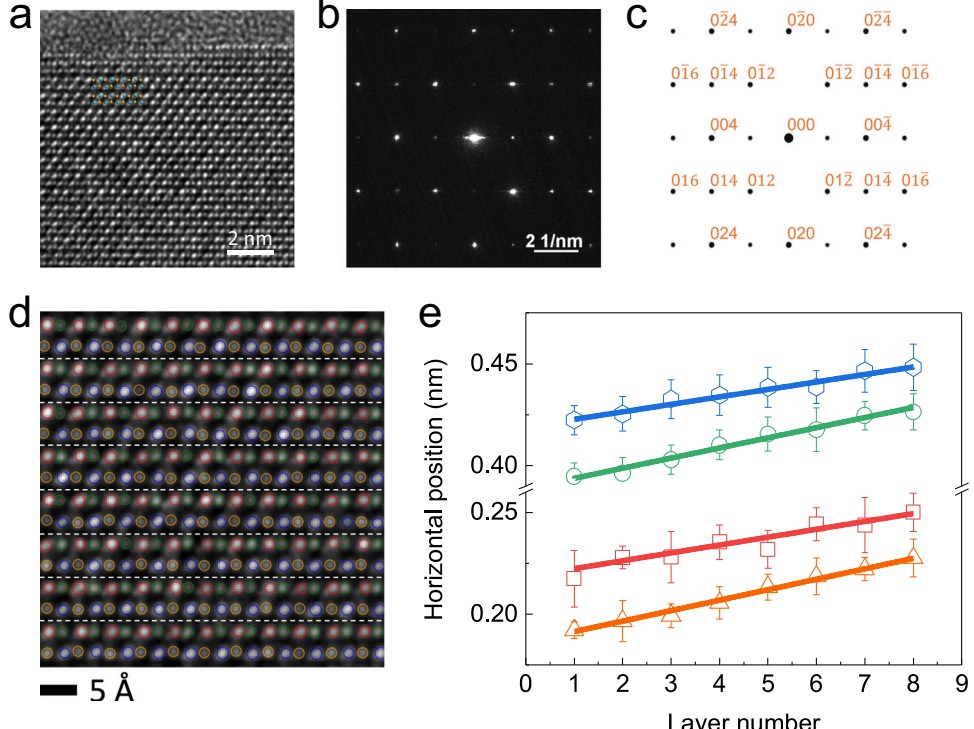

**Fig. 3 | TEM evidence of the AA stacking and interlayer sliding in SnSe. a** The (010) crystal plane of synthesized SnSe imaged by HRTEM. Heavier Sn atoms are brighter while the lighter Se atoms are slightly darker. A schematic crystal model of AA-stacked SnSe (inset) can fit well with our experimental data. The top amorphous region corresponds to the metal deposited during the cross section preparation process. SAED (selected area electron diffraction) patterns of SnSe along the [010] zone axis through the experiments of synthesized SnSe (**b**) and the simulations of bulk AA-SnSe (**c**). The perfect consistency between experiments and simulations further verifies the AA-stacking symmetry. **d** Analysis of the atomic positions from the side view imaged by HRTEM. Atoms are classified into four types in terms of their positions within the unit cell, marked by orange, red, green and blue spheres, respectively. **e** The horizontal position of atoms from different layer numbers. The error bars indicate the variance of individual atoms' horizontal position within each layer. A collective interlayer sliding is explicitly observed. From the linear fitting, the average atomic sliding is $0.052 \pm 0.002$, $0.039 \pm 0.005$, $0.050 \pm 0.003$, $0.037 \pm 0.003$ Å, respectively.

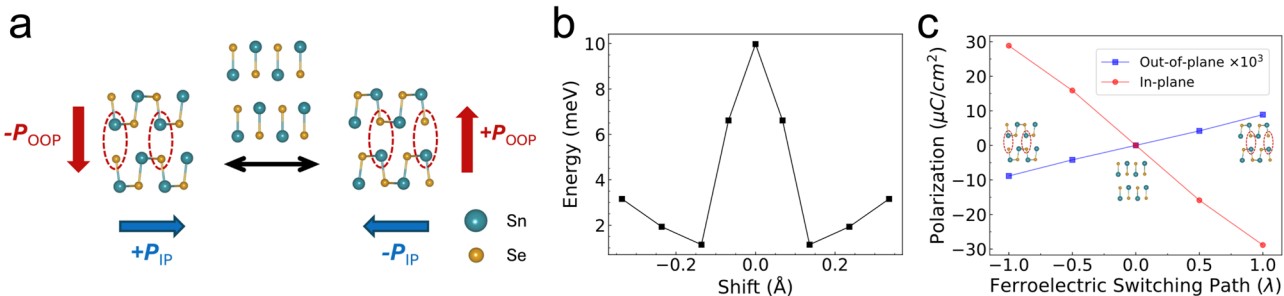

**Fig. 4 | The coupled IP and OOP ferroelectric switching mechanism in SnSe. a** Schematic illustration of the ferroelectric switching with coupled IP and OOP dipoles. The net IP ferroelectric polarization accumulates from the AA stacked layers. OOP ferroelectric polarization forms due to the sliding-induced misalignment of charged ions ($Sn^{2+}$ and $Se^{2-}$) from neighboring layers. The dominant dipole is denoted by the dashed ovals. When the IP ferroelectric configuration is switched (blue arrow), the OOP polarization (red arrow) switches simultaneously, and vice versa. **b** Calculated energy profile associated with the interlayer sliding (shift), showing a double-well-like behavior. **c** Calculated net OOP and IP polarization (per unit cell) along the ferroelectric switching path ($\lambda$ is the relative NEB coordinate). The OOP and IP polarizations are switched simultaneously resulting from the interlayer sliding configuration. The inset denotes the ferroelectric ground states and intermediate state as illustrated in (**a**).

the lowest-energy amount of interlayer sliding, both the IP and OOP polarization values are calculated along the ferroelectric switching path using the nudged elastic band (NEB) method (Fig. 4c), which reveals a coupled switching behavior, where the IP polarization and OOP polarization changes synchronously. The correlation between the IP and OOP switching is further supported by the modulation of IP optical SHG using an OOP E-field (Supplementary Fig. S12). The emergence of such an interlocked mechanism will provide extra tunability for future state-of-the-art device applications based on SnSe or similar ferroelectrics.

## Ferroelastic domain imaging

In addition to ferroelectricity, ferroelasticity also attracts concerted research interest and it has long been predicted in the MX family[19,20]. However, experimental demonstration of ferroelastic MXs is lacking, mainly due to technical challenges. HRTEM can detect the atomic displacement with a resolution of 0.5 Å[55]. Consequently, it is regarded as an optimal technique to directly image the ferroelastic domains. Pioneering theories pointed out that two characteristic ferroelastic domains, (100) and (010), coexist in SnSe nanosheets, and the ferroelastic phase transition occurs when the mechanical stress is applied

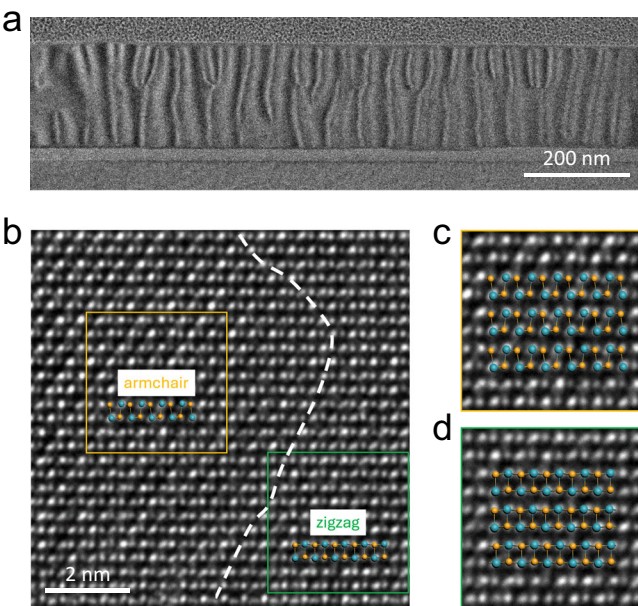

**Fig. 5 | Two types of ferroelastic domains imaged by HRTEM. a** Low-magnification cross-sectional TEM of SnSe. The stripes-like contrast variation indicates the presence of strain which is typically associated with ferroelastic domains. **b** Side-view HRTEM image under a higher magnification. A typical (010) type (armchair) configuration is shown in the yellow box and (100) type (zigzag) configuration is shown in the green box. The white dashed line indicates the domain boundary. **c, d** Zoom-in images of the atomic arrangements inside the two domains, showing good agreement with the (010) and (100) crystal plane.

along x or y crystal orientation[19,20] (Fig. 1d). Such stress distribution and the formation of ferroelastic domains is most evident from cross-sectional TEM images. From a low-magnification side view (Fig. 5a), the stark contrast in the image exhibits alternating bright and dark stripes-like patterns suggesting a non-uniform strain distribution, which is typical of ferroelastic materials[56,57]. With a higher-magnification image taken around the domain wall (Fig. 5b and Supplementary Fig. S13), we can clearly reveal the coexistence of (100) (green box) and (010) (yellow box) domains on the same cross section. The inset within the yellow (green) box schematically shows the fitting of Sn and Se positions, corresponding to an armchair (zigzag) type of structure. Note that the notation of armchair and zigzag here follows the convention used in previous anisotropic Raman studies[58], and is characteristic for the (010) and (100) planes of the puckered SnSe structure. We also utilize polarized Raman measurements from the top surface to distinguish the anisotropy between these two types of domains, and the results are summarized in Supplementary Fig. S14. Even higher-magnification images inside the yellow/green boxes are shown in Fig. 5c, d, confirming the good agreement of two types of ferroelastic domains, unambiguously proving the coexistence of two ferroelastic ground states in SnSe.

In summary, we experimentally report the coexisting ferroelectricity and ferroelasticity in MXs. Low-temperature (250 °C) synthesis of SnSe is highly amenable to the BEOL of semiconductor manufacturing process. We observed for the first time the interlayer sliding enabled OOP polarizations in MX systems. Correlated IP and OOP ferroelectric switching is identified in SnSe due to the unique stacking configuration. The switching voltage exhibits a nearly linear scaling with respect to the sample thickness, decreasing to as low as 0.28 V in a 5-nm-thick sheet. The addition of ferroelastic to MXs greatly enhances the tunability of ferroelectric properties potentially by strain engineering. The discovery of multiferroic behavior in layered SnSe presents an exciting landmark in the innovation of low-power and highly tunable memory[40] and logic devices[11,41].

## Methods

### Material synthesis

AA-stacking SnSe single crystals are directly synthesized on silicon by the low-temperature PVD method. Commercialized SnSe powders (99.999% metals basis, Sigma-Aldrich) are used as the source and a mixture of Ar/H$_2$ gas is chosen as the carrier gas. Typical growth conditions utilize low pressure of 60 mTorr and flow rate of 60 standard cubic centimeters per minute (sccm). The source temperature at the center of the tube furnace is set to 600–650 °C, and the evaporated SnSe is deposited downstream onto the substrate at a low temperature of 250 °C. To favor the formation of thinner flakes, the center temperature is reduced to 550–600 °C, with increased flow rate of 100 sccm and pressure of 150 mTorr, along with reduced growth time. The typical lateral size for 5–10 nm thick flakes is 5–10 μm, and 10–20 μm for relatively thicker flakes. The uniformity and yield of the AA-stacking phase are verified by HRTEM and electron diffraction across multiple samples.

### SHG measurements and calculations

Optical SHG signals of SnSe nanoflakes are measured under both 800 nm femtosecond laser (main text) and 1064 nm CW laser (Supplementary Information) excitation. The 800 nm fs measurements are performed in a home-built setup using a Ti:sapphire laser (Coherent Chameleon Ultra II, 140 fs, 80 MHz), focused onto a 1–2 μm spot with a 50x objective lens. The polarization state was established with a linear polarizer and half-wave plate, while the second harmonic beam was picked by a shortpass filter and analyzed by another half-wave plate and linear polarizer. CW 1064 nm measurements are performed using a Horiba Jobin Yvon LabRAM ARAMIS confocal Raman microscope. The linear polarizer and analyzer are parallel/orthogonal to each other while the samples are rotated to probe the polarization dependence. The CW laser (Laser Quantum Ventus 1064) is focused onto a ~1 μm spot with a 100x objective. The calculation of angle-dependent SHG spectra is obtained using the ArchNLO package based on the perturbation theory of polarization operator. A dense k-grid of $27 \times 27 \times 1$, and the inclusion of 200 conduction bands are used to get the converged optical susceptibilities.

### PFM measurements

DRAT PFM (MFP-3D, Asylum Research) is employed to explore the ferroelectric switching of SnSe in both horizontal and vertical modes. A metallic Pt-coated cantilever (MikroMasch HQ: NSC18/PT) is used to detect the local piezoelectric deformation of sample. Under the horizontal mode, the electric field forms between the tip and side electrode. As for the vertical mode, the electric field lies between the tip and bottom p+ Si substrate. The data was recorded in the non-volatile "DC off" mode, while the "DC on" data is shown in Supplementary Fig. S4 for comparison.

### TEM characterization

Both the top and side view of SnSe nanoflakes are captured by HRTEM. Top-view SnSe nanoflakes are directly transferred onto commercialized TEM grids via dry transfer method. The side-view samples are fabricated by focused ion beam (FIB). At the end of FIB process, 5 kV Ga ion beam is utilized to remove the damage of samples by 30 kV Ga ion beams. After the delicate sample fabrication, the HRTEM images and electron diffraction patterns are taken by using a FEI Titan operated at 200 kV and equipped with an image Spherical Aberration Corrector at 200 kV.

### First-principles calculations

The ground-state structural and electronic properties of bilayer SnSe are calculated by DFT within the generalized gradient approximation (GGA) using the Perdew-Burke-Ernzerhof (PBE) functional, as implemented in the Vienna ab initio simulation package (VASP). A k-point sampling of $12 \times 12 \times 1$ and a plane-wave basis with an energy cutoff of

450 eV is employed to obtain the converged charge densities and wavefunctions. The vdW interactions are included via the semi-empirical Grimme-D3 scheme with zero-damping. A vacuum distance of 18 Å between adjacent layers is used along the periodic direction to avoid spurious interactions. Spin orbit coupling (SOC) is included. The energy barrier of the ferroelectric phase transition from negative to positive OOP polarized configuration is calculated by the nudged elastic band (NEB) method.

## Reporting summary

Further information on research design is available in the Nature Portfolio Reporting Summary linked to this article.

## Data availability

Relevant data generated in this study are provided in the article and Supplementary Information. All raw data that support the plots within this paper and other findings of this study are available from the corresponding authors upon request.

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

## Acknowledgements

R.C., F.M. and J.Y. acknowledge the support by Intel Corporation under an award titled Valleytronics center, and the support from the U.S. Department of Energy, Office of Basic Energy Sciences, Materials Sciences and Engineering Division, under Contract DE-AC02-05-CH11231 (Organic–Inorganic Nanocomposites KC3104). R.R. acknowledges the Air Force Office of Scientific Research 2D Materials and Devices Research program through Clarkson Aerospace Corp under Grant No. FA9550-21-1-0460. X.H. is supported by the SRC-ASCENT center which is part of the SRC-JUMP program. Use of the Center for Nanoscale Materials, an Office of Science user facility, was supported by the U.S. Department of Energy, Office of Science, Office of Basic Energy Sciences, under Contract No. DE-AC02-06CH11357. Work at the Molecular Foundry was supported by the Office of Science, Office of Basic Energy Sciences, of the US Department of Energy under contract no. DE-AC02-05CH11231. The devices were fabricated in the UC Berkeley Marvell Nanofabrication Laboratory. We acknowledge the Biomolecular Nanotechnology Center for access and assistance with measurement systems.

## Author contributions

R.C., F.M. and J.Y. conceived the project and designed the experiments. R.C. performed the PFM measurements. Y.Z.L., T.Z., R.C., F.M. and Y.L. conducted the TEM characterizations and analyses. X.X., L.Z. and L.Y. carried the theoretical calculations. F.M., R.C., J.C., F.Y. and E.C. obtained the optical data. R.C., J.C., S.Y. and J.Z. synthesized the samples. R.C., S.Y., H.Z., X.H., X.C., T.C.Z., Y.C. and K.D. fabricated the devices. R.C., F.M., H.Z., P.C. and Y.C. performed electrical measurements under the supervision of J.W. K.W. and T.T. grew the hBN crystal. C.-C.L., A.V.P., I.Y. and R.R. contributed to the data analysis. R.C., F.M. and J.Y. wrote the manuscript. All authors discussed the results and commented on the manuscript.

## Competing interests

The authors declare no competing interests.

## Additional information

¹Department of Materials Science and Engineering, University of California, Berkeley, CA 94720, USA. ²Materials Sciences Division, Lawrence Berkeley National Lab, Berkeley, CA 94720, USA. ³Ningbo Institute of Materials Technology & Engineering, Chinese Academy of Sciences, Ningbo 315201, China. ⁴Center for Nanoscale Materials, Nanoscience and Technology Division, Argonne National Laboratory, Lemont, IL 60439, USA. ⁵College of Industry-Education Integration, Nanjing University of Posts and Telecommunications, Nanjing 210023, China. ⁶Department of Physics and Institute of Materials Science and Engineering, Washington University in St. Louis, St. Louis, MO 63130, USA. ⁷Department of Materials Science and Engineering, North Carolina State University, Raleigh, NC 27606, USA. ⁸National Institute for Material Science, Tsukuba 305-0047, Japan. ⁹Components Research, Intel Corporation, Hillsboro, OR 97124, USA. ¹⁰The Molecular Foundry, Lawrence Berkeley National Laboratory, Berkeley, CA 94720, USA. ¹¹Department of Physics, University of California, Berkeley, CA 94720, USA. ¹²These authors contributed equally: Rui Chen, Fanhao Meng, Hongrui Zhang, Yuzi Liu, Shancheng Yan. ✉e-mail: fhmeng@berkeley.edu; yaojie@berkeley.edu

