## [Transparent Peer Review file · Nature Communications]

Room-temperature multiferroicity in sliding van der Waals semiconductors with sub-0.3 V switching

Corresponding Author: Professor Jie Yao

Version 0:

Reviewer comments:

Reviewer #1

(Remarks to the Author)

The authors have presented a multiferroic phase of tin selenide (SnSe), wherein the inversion symmetry breaking is sustained in AA-stacked multilayers across a broad range of thicknesses. Thickness scaling yields a sub-0.3V ferroelectric switching, which holds promise for future low-power-consumption applications. Coexisting armchair- and zigzag-like structural domains have been imaged under electron microscopy, providing experimental evidence for the degenerate ferroelastic ground states that have been theoretically predicted. A major revision is recommended before it can be considered for acceptance. Specific comments are provided below.

1. The material thickness scales down, the ferroelectric switching voltage of SnSe reaches sub-0.3 V, which offers new opportunity towards future attojoule energy-efficient applications. However, SnSe single crystals is grown on a variety of substrates, including silicon. Is there a corresponding lattice mismatch between its stacking structure and the substrate? This mismatch strain might also cause a change in the properties of the ferroelectric stacked vdw layer, especially when the layer thickness is reduced to 5nm layer thickness.
2. The authors claim that even sub-100 mV ferroelectric switching could potentially be achieved in fewer layers of SnSe nanosheets with future efforts. This would require a reduction of several orders of magnitude from the existing 0.28V. However, the thickness is already at 5nm, what is the basis for this conclusion?
3. Based on the linear fitting, The author obtained a constant and small coercive field of 198 kV/cm. The author should add more evidence for 198KV/cm as a small coercive field.
4. The main consideration in the DFT calculation process is the interlayer slip. A clear two-dimensional displacement behaviour is shown in Figure 4a. However, SnSe is a three-dimensional structure. Is there also misalignment or rotation between the layers and can this misalignment or rotation also cause asymmetry or interlayer dipoles in the structure?
5. From a low magnification side view (Fig. 5a), the strong contrast in the image shows alternating bright and dark stripe-like patterns, suggesting a non-uniform strain distribution. The domain wall region is shown in Figure 5b. Please indicate the position of the region in Fig. 5b in Fig. 5a.
6. From the non-uniform strain distribution in Figure 5a, is it possible to speculate that its stripe-like pattern actually contains richer physics? For example, the author claims that the non-uniform strain distribution is due to ferroelasticity. Then the following conditions must be ruled out: the mechanical mismatch generates between the top and bottom surfaces during the preparation process, leading to the non-uniform deformation of the intermediate structure, which ultimately creates the stripe-like pattern? Also, is it possible that flexural electrical properties may also occur with such a large deformation inhomogeneity?
7. Fig. 2b does not seem to be described or cited in the text. In comparing Figures 2b and d, the description of the sample thickness in Figure 2b is missing.
8. Progressively reducing the thickness will enable sub-100 mV ferroelectric switching. Please briefly describe the extrapolation formula or the basis for the calculation.

Reviewer #2

(Remarks to the Author)

In this manuscript, the authors report the multiferroicity in layered SnSe at room temperature, supported by comprehensive experimental characterizations and theoretical calculations. The ferroelectric phase of SnSe is confirmed by both optical SHG measurement and PFM, and the capability to switch the material both in-plane and out-of-plane is demonstrated.

Moreover, the stacking and sliding between the van der Waals layers are analyzed by HRTEM, and two types of ferroelastic domains are also imaged. The discovery of interlayer sliding which induces coupled ferroelectric behavior is particularly interesting to me, and the discussion on sliding ferroelectricity is of current interest to the community. Considering the wide interest in 2D ferroic materials and sufficient novelty in this work, I believe it is worth publishing in Nature Communications after the following concerns are well addressed.

1. The ferroelectric (FE) and anti-ferroelectric (AFE) stacking of this family of material has been theoretically studied, where both AA and AC stacking are considered FE type. The authors compared AA versus AB stacking with SHG and SAED, but does the diffraction pattern also rule out AC stacking? More detailed comparison should be provided.
2. The interlayer sliding is well presented in Figure 3, but the mechanism where sliding introduces OOP ferroelectricity (mainly discussed in page 6) is a bit vague. Does the OOP polarization come from interlayer charge transfer or a structural transformation? And what's the exact role of in-plane sliding direction (line 15) here? The authors can elaborate more on the explanation.
3. The authors construct a bilayer model to perform DFT calculations, while in the TEM images, all the layers are collectively sliding, where it's not appropriate to assume periodic boundary condition. Can the authors discuss the trend when more than two layers are accounted into the sliding?
4. In terms of the armchair and zigzag domains in Figure 5, due to the anisotropy of this orthorhombic crystal, it could be interesting to see if there is any optical contrast between the domains. If the domains are large enough, optical characterizations are more convenient than relying on TEM.
5. Minor suggestions: the figures can be further polished to facilitate the reading. In Figure 2 and 4, the labels are quite small and the fonts are inconsistent; some panels (such as Figure 5a) are not very clear and the resolution or contrast can be improved.

Reviewer #3

(Remarks to the Author)

This work explores multiferroic phases of tin selenide (SnSe), where the inversion symmetry breaking is maintained in AA stacked multilayers over a range of thicknesses. This structure results in out of plane ferroelectric behavior, in addition to the previously observed in-plane polarization. The key point is that the observed OOP ferroelectric response offers a path to active devices and the voltage dependence observed suggests a potential for low-voltage operation of those devices enabled.

This work would be greatly improved if the authors focus more directly on the OOP polarization phase induced and the materials development steps needed to result in these phases. It is not clear from the text how easily or how large the AA polarized films can be produced. This needs to be more clearly addressed. The formation of this homolog is key to device applications particularly low voltage operation.

The OOP behavior is the real "new" result here. It needs to be focused on in more detail than exists in the current manuscript. The formation of the AA state is key. It's not clear from the manuscript if AA is strain induced or naturally occurring. Do the author randomly find AA platelets created in their growth process? Are the AA stacks uniform and stable?

They show low, sub -0.3V ferroelectric switching, important for low-power consumption applications. In addition, this work also reports how stacking controls both IP and OOP ferroelectric properties and the ability to couple these polar interactions.

This manuscript needs revision to focus more directly on the viability of creating AA stacked materials for device applications, again the OOP performance and the low voltage operation are the novel findings in this work. This refocusing is needed before this work adequate for publication.

Version 1:

Reviewer comments:

Reviewer #1

(Remarks to the Author)

The author answered my questions and supplemented relevant data.

Reviewer #2

(Remarks to the Author)

In the revised version, the authors have provided sufficient experimental characterizations and theoretical analyses to address the comments raised by the reviewer. The questions are well explained, and the manuscript has been improved accordingly. I thus recommend its acceptance to Nature Communications.

Before publication, the authors may check the following minor suggestions:

1. The four coupled polarization states and the associated symmetry operations are thoroughly discussed in the responses (2nd question). The authors should include some of these descriptions into the SI as well.
2. The format of the images should be kept as consistent as possible, like the scale bar in TEM images, should be preserved as uniformly as possible.

Reviewer #3

(Remarks to the Author)

I believe the authors have addressed my primary concerns. Details on creation of the OOP polarization phase have been included. The authors state that they have included uniformity and yield information on the OOP phase in the response, but I failed to identify that language in the main manuscript. It's still not clear to me if they can efficiently synthesize that phase for device applications.

The work is of significance to the field. I believe it should be accepted for publication.

Reviewer #1 (Remarks to the Author):

The authors have presented a multiferroic phase of tin selenide (SnSe), wherein the inversion symmetry breaking is sustained in AA-stacked multilayers across a broad range of thicknesses. Thickness scaling yields a sub-0.3V ferroelectric switching, which holds promise for future low-power-consumption applications. Coexisting armchair- and zigzag-like structural domains have been imaged under electron microscopy, providing experimental evidence for the degenerate ferroelastic ground states that have been theoretically predicted. A major revision is recommended before it can be considered for acceptance. Specific comments are provided below.

Authors' response:

We appreciate the reviewer's positive evaluations of our manuscript. Following the reviewer's comments, we have addressed all the issues in the revised manuscript, and point-by-point responses are listed below. We hope that with the revised manuscript, this work meets the publishing criteria of *Nature Communications*.

1. The material thickness scales down, the ferroelectric switching voltage of SnSe reaches sub-0.3 V, which offers new opportunity towards future attojoule energy-efficient applications. However, SnSe single crystals is grown on on a variety of substrates, including silicon. Is there a corresponding lattice mismatch between its stacking structure and the substrate? This mismatch strain might also cause a change in the properties of the ferroelectric stacked vdw layer, especially when the layer thickness is reduced to 5nm layer thickness.

Authors' response:

We thank the reviewer for the comments on the potential influence of lattice mismatch to the samples' stacking. The lattice mismatch between SnSe and silicon is around 10%, but such mismatch doesn't cause a significant change in the ferroelectric stacking, mainly because of the van der Waals nature of layered SnSe. First, the van der Waals interactions between the sample and its substrate is typically 1-2 orders of magnitude weaker than covalent bonds. The SnSe growth process is less impacted by such weak coupling with the substrate and thus, the mismatch induced strain is smaller. In our experiments, the growth is considered non-epitaxial, which is different from the conventional oxide thin film epitaxy. As can be seen from the optical image below (Fig. S1c, Fig. R1a), the crystalline orientations among different flakes are not aligned, suggesting a non-epitaxial condition. Similar results were observed on other substrates we tried. Secondly, although the sample-to-substrate coupling is weak, it still poses a certain amount of strain to the atomic layers that are directly in contact with the substrate. However, due to the interlayer van der Waals

coupling in the SnSe flakes, such effects are released within just the bottom few layers, and the rest of the layers are not involved. To illustrate this, we take the cross-section view (Fig. R1b) around the sample/substrate interface with HRTEM, where the stacking is consistent among almost all the layers, only except the bottom 1-2 layers.

Figure R1 | **a**, Optical image of SnSe nanosheets on silicon substrate. **b**, Cross-section view with HRTEM.

2. The authors claim that even sub-100 mV ferroelectric switching could potentially be achieved in fewer layers of SnSe nanosheets with future efforts. This would require a reduction of several orders of magnitude from the existing 0.28V. However, the thickness is already at 5nm, what is the basis for this conclusion?

Authors' response:

We thank the reviewer for raising this concern. We want to clarify that this sentence was intended to point out the future potential of ~ 0.1 V switching using thinner SnSe flakes. Apart from thinning down the material, strategies including interface engineering, defect engineering, doping engineering and improving contact could also assist in the reduction of switching voltage. The sample thickness of 5nm corresponds to roughly 10 layers of SnSe, and based on the observation that the switching voltage scales almost linearly with the sample thickness (Fig. 2f), we simply assumed that further thinning down to 3-4 layers would potentially enable ultra-low-voltage switching. However, thanks to the reviewer's question, we realized that this extrapolation is not rigorous (more discussion in question 3 and question 8 below) and might cause

confusion, so we removed the claim of sub-100 mV switching and made the following changes in the main text:

[Page 5, Line 27] Therefore, even sub-100 mV ferroelectric switching can potentially be achieved. Future efforts including interface engineering, defect engineering and doping engineering could potentially lead to ultralow voltage switching in fewer layer SnSe nanosheets.

[Page 8, Line 4] The switching voltage exhibits a nearly linear scaling with respect to the sample thickness, decreasing to as low as 0.28 V in a 5-nm-thick sheet. Progressively reducing the thickness will enable sub-100 mV ferroelectric switching.

3. Based on the linear fitting, The author obtained a constant and small coercive field of 198 kV/cm. The author should add more evidence for 198KV/cm as a small coercive field.

Table R1. Reported coercive fields of typical 2D van der Waals or thin-film ferroelectrics

Material	Coercive field (kV/cm)	Reference
CuInP ₂ S ₆	760 (30 nm)	Liu, F. et al. ¹
CuCrP ₂ S ₆	560 (25 nm)	Io, W. F. et al. ²
In ₂ Se ₃	309 (20 nm)	Zhou, Y. et al. ³
WTe ₂	500	Fei, Z. et al. ⁴
BaTiO ₃	10	Jiang, Y. et al. ⁵
Hf(Zr) _{1+x} O ₂	650	Wang, Y. et al. ⁶
Bi _{1.8} Sm _{0.2} O ₃	7000 (4.56 nm)	Yang, Q. et al. ⁷
SnSe	206 (35 nm)	This work

Authors' response:

We thank the reviewer for pointing out this issue. As listed in Table R1, we briefly summarize the coercive fields in previous reports of typical 2D vdW ferroelectrics and oxide thin films. We conclude that ~200 kV/cm is relatively small among similar families of materials. We also admit that when comparing with the most recent report in ultrathin BaTiO₃⁵, the coercive field of 2D vdW ferroelectrics still faces a gap. In addition, we noticed that our original expression could be misleading because the coercive field does not directly correspond to the slope of linear fitting, so we make the following modifications in the main text:

[Page 5, Line 23] The linear fitting gives out a slope of about 200 kV/cm, which agree with the nearly constant coercive field in samples above 35 nm (Supplementary Fig. S6). As the material further thins down,

more reduction of the switching voltage is expected, although the measured coercive field tends to increase, which might pose a limit to the smallest switching voltage we can achieve.

4. The main consideration in the DFT calculation process is the interlayer slip. A clear two-dimensional displacement behaviour is shown in Figure 4a. However, SnSe is a three-dimensional structure. Is there also misalignment or rotation between the layers and can this misalignment or rotation also cause asymmetry or interlayer dipoles in the structure?

Authors' response:

We thank the reviewer for pointing out other possibilities of interlayer structural configuration that could cause symmetry breaking and electric dipoles. Indeed, misalignment between the vdW layers can break the inversion symmetry and cause interlayer dipoles, in a similar fashion as the interlayer sliding we observed. However, the sliding we report here is a collective behavior among the majority of the layers (Fig. 3d-e), while the misalignment is rarely found and considered as stacking fault in this case. In Fig. R2a, we show an example of interlayer misalignment observed in one of the HRTEM cross-section images, but the rest of the layers are consistently aligned. We also provide another cross-section image of (010) plane and follow the same method shown in Fig. 3d-e to demonstrate that the sliding is a global behavior (Fig. R3).

Figure R2 | **a**, Cross-sectional HRTEM image. Arrow marks the position where misalignment or stacking fault observed. **b**, Selected area electron diffraction from [001] direction, indicating a single crystal with no interlayer twist.

Figure R3 | TEM analysis of the interlayer sliding in an additional sample image. a, Analysis of the atomic positions from the side view imaged by HRTEM. Atoms are classified and marked by the same color scheme as in Fig. 3d. **b**, The horizontal position of atoms from different layers, confirming a collective interlayer sliding.

The reviewer also mentioned the possibility of rotation, which has been reported in twisted TMDC and other systems⁸ to induce interlayer dipoles. Such twisting behavior can be verified from the electron diffraction pattern perpendicular to the sample surface. In Fig. R2b, our SAED result confirms that there are no satellite peaks in the diffraction pattern, which rules out the twist-induced ferroelectricity in our system.

We have added Fig. R3 and Fig. R2b to *Supplementary Fig. S9 and Fig. S8a*.

5. From a low magnification side view (Fig. 5a), the strong contrast in the image shows alternating bright and dark stripe-like patterns, suggesting a non-uniform strain distribution. The domain wall region is shown in Figure 5b. Please indicate the position of the region in Fig. 5b in Fig. 5a.

Authors' response:

We thank the reviewer for pointing out this issue in our figures. We apologize for the fact that we didn't keep track of the exact location where the high-magnification image Fig. 5b was taken in reference to the low-magnification image Fig. 5a. To demonstrate the correspondence between the stripes and the domain wall, we show another set of data where the imaged spot are recorded in detail, as shown in Fig. R4. These results are also added into the *supplementary information (Fig. S13)*.

Figure R4 | Ferroelastic domains imaged by TEM. *a*, Low-magnification side view showing stripes-like contrast. The white box marks the region where the HRTEM image in panel *b* is taken. *b*, HRTEM image of the selected region under higher magnification. The white dashed line indicates the domain boundary.

6. From the non-uniform strain distribution in Figure 5a, is it possible to speculate that its stripe-like pattern actually contains richer physics? For example, the author claims that the non-uniform strain distribution is due to ferroelasticity. Then the following conditions must be ruled out: the mechanical mismatch generates between the top and bottom surfaces during the preparation process, leading to the non-uniform deformation of the intermediate structure, which ultimately creates the stripe-like pattern? Also, is it possible that flexural electrical properties may also occur with such a large deformation inhomogeneity?

Authors' response:

We thank the reviewer for the insightful comments on the stripe-like patterns and the physics behind. We will address this question from three aspects as follows:

(1) First, we totally agree with the reviewer that there could be richer physics underneath such non-uniform strain distribution. One natural possibility is to look into the domain boundaries, and to analyze the local polarization states around the domain wall region. Also, the in-situ study of these domains under external mechanical/electrical stimulation, or through temperature cycling, would be interesting topics. Furthermore, considering that SnSe is a semiconductor, the charge carriers could be driven and re-distribute under such strain inhomogeneity. The charge redistribution in such complex ferroelastic strain fields, combined with the ferroelectric polarization within the material, is worth dedicated investigation. We believe these topics

are beyond the scope of our current work, and deserve to be established as separate projects for future exploration, so we didn't include such discussions in this manuscript.

(2) Secondly, we believe this observation of non-uniform strain is internal, instead of being induced by external factors during sample preparation. The cross-section samples were prepared by Gallium FIB. The rough milling was performed at 30kV until the lamella was thinned down to about 300nm. Then, the sample was cleaned by using 5kV Ga beam to remove the damage introduced by 30kV Ga beam. Before the sample was loaded to TEM column, the sample was milled by 1kV Argon ion beam. Based on our experience, the damage or deformation to the sample from preparation is minimized by following this procedure, which has been verified in many samples we prepared for other experiments. Also, from the morphology of the area above and below the SnSe region, there is no obvious wrinkles observed. Thus we believe the feature is native to the sample itself.

(3) Thirdly, we agree with the reviewer's assumption that flexoelectric effect may occur under such strain gradient. Generally speaking, when 2D materials or thin films scale down to nanometer thicknesses, the strain gradient becomes much more significant and leads to pronounced flexoelectric response. Also, for ferroelectrics as typical high-k materials, it's valid to expect strong flexoelectricity. Following the reviewer's suggestion, we notice that there are already some theoretical calculations focusing on the flexoelectric effect in 2D materials such as transition metal dichalcogenides, phosphorene and group-IV monochalcogenides⁹⁻¹¹. We believe such investigation is also valuable to our system, but will add more complexity and distraction to the main focus of this manuscript. We plan to incorporate these valuable topics in future projects.

7. Fig. 2b does not seem to be described or cited in the text. In comparing Figures 2b and d, the description of the sample thickness in Figure 2b is missing.

Authors' response:

We thank the reviewer for noticing this omission we made. And we have corrected this in the main text:

[Page 4, Line 32] As shown in Fig. 2b, the typical butterfly-shaped amplitude behavior and the hysteretic phase against bias curve indicate the in-plane ferroelectricity in the SnSe sample.

[Page 10, Line 6] Fig. 2 ... The sample thickness is 81.6 nm.

8. Progressively reducing the thickness will enable sub-100 mV ferroelectric switching. Please briefly describe the extrapolation formula or the basis for the calculation.

Authors' response:

We thank the reviewer for helping us realize this issue. As we discussed in question 2, this conclusion was deduced from the linear thickness scaling of the switching voltage. To avoid any ambiguity, we have changed the original sentences in the main text, and removed the claim of sub-100 mV switching:

[Page 5, Line 23] The linear fitting gives out a slope of about 200 kV/cm, which agree with the nearly constant coercive field in samples above 35 nm (Supplementary Fig. S6). As the material further thins down, more reduction of the switching voltage is expected, although the measured coercive field tends to increase, which might pose a limit to the smallest switching voltage we can achieve. ~~Therefore, even sub-100 mV ferroelectric switching can potentially be achieved~~ Future efforts including interface engineering, defect engineering and doping engineering could potentially lead to ultralow voltage switching in fewer layer SnSe nanosheets.

[Page 8, Line 4] The switching voltage exhibits a nearly linear scaling with respect to the sample thickness, decreasing to as low as 0.28 V in a 5-nm-thick sheet. ~~Progressively reducing the thickness will enable sub-100 mV ferroelectric switching.~~

In terms of the extrapolation method in the low voltage/ultrathin limit, it is necessary to analyze the evolution of the coercive field (E_c) itself as the sample thins down. It can be seen from Fig. R5a that, the coercive field is almost constant of ~ 206 kV/cm above 35 nm thick samples. As the thickness further goes down, E_c scales up, similar to the classical Janovec-Kay-Dunn (JKD) scaling. From the log plot in Fig. R5b, the slope is estimated to be -0.476, which is smaller than the conventional JKD scaling ($E_c \propto d^{-2/3}$). The sub-JKD scaling behavior is often associated with the presence of depolarization fields, and might also be related to the van der Waals structure of 2D ferroelectrics, which is worth future exploration.

Figure R5 | Plot of the measured coercive fields with respect to sample thicknesses. Red dashed line: fitting of the data points below 35 nm thickness, where the coercive field starts to increase with reduced thickness. The slope of -0.476 from log plot indicates a sub-JKD scaling behavior. Blue dashed line: fitting of the data points above 35 nm thickness, where the coercive field is roughly thickness independent.

We add these plots and discussion of the coercive field curve in *Supplementary Fig. S6*.

Reviewer #2 (Remarks to the Author):

In this manuscript, the authors report the multiferroicity in layered SnSe at room temperature, supported by comprehensive experimental characterizations and theoretical calculations. The ferroelectric phase of SnSe is confirmed by both optical SHG measurement and PFM, and the capability to switch the material both in-plane and out-of-plane is demonstrated. Moreover, the stacking and sliding between the van der Waals layers are analyzed by HRTEM, and two types of ferroelastic domains are also imaged. The discovery of interlayer sliding which induces coupled ferroelectric behavior is particularly interesting to me, and the discussion on sliding ferroelectricity is of current interest to the community. Considering the wide interest in 2D ferroic materials and sufficient novelty in this work, I believe it is worth publishing in Nature Communications after the following concerns are well addressed.

Authors' response:

We appreciate the reviewer's positive evaluation of our manuscript. We have addressed all the comments point by point and revised the manuscript accordingly.

1. The ferroelectric (FE) and anti-ferroelectric (AFE) stacking of this family of material has been theoretically studied, where both AA and AC stacking are considered FE type. The authors compared AA versus AB stacking with SHG and SAED, but does the diffraction pattern also rule out AC stacking? More detailed comparison should be provided.

Authors' response:

We thank the reviewer for raising the possibility of other stacking configurations. To confirm the AA stacking of our SnSe sample and distinguish it from AC stacking, we conduct electron diffraction from the out-of-plane direction, [001], and the results are summarized in Fig. R6. The simulated diffraction patterns for AA (Fig. R6b) and AC (Fig. R6c) stacking are different, and can be distinguished by measuring the d spacing of the nearest peak spot. The experimental SAED result shows good consistency with AA stacking.

We have added these plots and comparison in Supplementary Fig. S8.

Figure R6 | Selected area electron diffraction from [001] direction. a, Experimental result. **b**, Simulation of AA stacking, the d spacing matches with experimental results perfectly. **c**, Simulation of AC stacking.

2. The interlayer sliding is well presented in Figure 3, but the mechanism where sliding introduces OOP ferroelectricity (mainly discussed in page 6) is a bit vague. Does the OOP polarization come from interlayer charge transfer or a structural transformation? And what's the exact role of in-plane sliding direction (line 15) here? The authors can elaborate more on the explanation.

Authors' response:

We thank the reviewer's constructive comments, and we apologize that our original explanation might not be clear enough. In our model (Fig. 4), the OOP polarization forms when the Sn and Se atoms from neighboring layers are misaligned, due to the interlayer sliding. Our DFT calculations show that although such interlayer shift might cause certain changes in the electronic band structure and thus induce charge transfer, the dominant factor is still the imbalanced electric dipoles directly resulting from the sliding structure, so the OOP ferroelectricity here comes from the structural transformation.

We clarify that the in-plane sliding direction refers to the leftward/rightward sliding when viewing from the [010] direction. For example, in Fig. 4a, the top layer is sliding leftward with respect to the bottom layer. We add more comprehensive notations in Fig. R7 below, where the in-plane sliding direction is either +x or -x. It's worth noting that the sliding is a collective behavior experimentally observed among all the layers, and its direction is assumed to be fixed in our model, otherwise the energy barrier of shifting the overall sliding direction would be too high. In this case, the sliding direction determines how the IP and OOP polarization are coupled. For example, if the sliding direction is -x (state A and B in Fig. R7), then the IP polarization $-P_x$ is locked with OOP polarization $+P_z$ (state A), and vice versa (state B). From the symmetry point of view, such IP and OOP coupling are described by a C_{2y} operation, which always flips both IP and OOP polarizations simultaneously. The role of sliding direction can be described by a C_{2x} operation, which preserves the IP polarization but flips the OOP one, which is why in the main text we describe the OOP polarization as “*determined by the IP sliding direction*”.

Thanks to the reviewer's comments, we made the following changes in the main text to illustrate the OOP switching mechanism more clearly:

[Page 6, Line 20] The direction of OOP polarization is determined by the sliding direction and locked to the IP polarization state, thus accounts for the observed OOP ferroelectricity. With a leftward sliding as in Fig. 4a, when the IP state is switched from $+P_{IP}$ to $-P_{IP}$, the OOP dipoles flip from $-P_{OOP}$ to $+P_{OOP}$ correspondingly, and vice versa. Such coupling arises as a direct consequence of AA stacking and sliding structure, as the $(-P_{OOP}, +P_{IP})$ and $(+P_{OOP}, -P_{IP})$ ground states are intrinsically associated with a C_{2y} symmetry operation in this fashion (Supplementary Fig. S10).

And we add the schematic illustration to *Supplementary Fig. S10*:

Figure R7 | Four energy-equivalent polarization states considering different combinations of sliding direction and in-plane polarization direction. The arrows mark the correlation between the states and the corresponding symmetry operations.

3. The authors construct a bilayer model to perform DFT calculations, while in the TEM images, all the layers are collectively sliding, where it's not appropriate to assume periodic boundary condition. Can the authors discuss the trend when more than two layers are accounted into the sliding?

Authors' response:

We thank the reviewer's kind suggestions. To further demonstrate the multilayer collective sliding, we perform DFT calculation to a 3-layer structure as shown in Fig. R8. The results are summarized in the table below. The stabilized structure still shows considerable amount of sliding among the three layers, and the atomic shifts are on the same order of magnitude as experimentally observed. In all our DFT calculations, to avoid spurious interactions between periodic images, a vacuum distance of at least 15 Å is introduced in the z direction. The initial values for the atomic shifts are adequately larger than the final stabilized value to avoid artifacts. All lattice parameters, atomic positions, and structures are fully optimized until a stable final state was achieved.

We add these plots and discussion in *Supplementary Fig. S11*:

Figure R8 | Calculated atomic shift for the 4 types of atoms within a unit cell (3-layer model).

4. In terms of the armchair and zigzag domains in Figure 5, due to the anisotropy of this orthorhombic crystal, it could be interesting to see if there is any optical contrast between the domains. If the domains are large enough, optical characterizations are more convenient than relying on TEM.

Authors' response:

We thank the reviewer for the valuable suggestion. SnSe indeed exhibits large optical anisotropy, and the armchair and zigzag orientations can be distinguished by polarization resolved Raman. Here we use polarized Raman mapping to search for in-plane domains that are large enough for optical characterizations, and the result is shown in Fig. R9. With both the polarizer and analyzer horizontal (parallel setting), we locate a $\sim 3 \times 3 \mu\text{m}$ region with a distinguishable domain boundary (Fig. R9a, Raman A_g^2 peak intensity map). Upon rotating the polarization from 0° to 90° , the intensity contrast between domain A and domain B flips (Fig. R9b). The full Raman spectra from the two domains under horizontal/vertical polarizations are shown in Fig. R9c-d, clearly exhibiting the discrepancy due to crystalline anisotropy. In addition, we pick a spot in each domain and conduct a full cycle of angle-dependent measurement respectively. The polar plots for the 3 representative Raman peaks (B_{3g} , A_g^2 and A_g^3) are shown in Fig. R9e-g. It is evident that the lattice orientations of the two domains are $\sim 87^\circ$ apart, one corresponding to the armchair-like and the other zigzag-like.

We make the following changes in the manuscript and add these figures and descriptions in *Supplementary Fig. S14*.

[Page 7, Line 21] Note that the notation of armchair and zigzag here follows the convention used in previous anisotropic Raman studies, and is characteristic for the (010) and (100) planes of the puckered SnSe structure. We also utilize polarized Raman measurements from the top surface to distinguish the anisotropy between these two types of domains, and the results are summarized in Supplementary Fig. S14.

Figure R9 | Linearly polarized Raman mapping with 532 nm excitation. **a-b**, Intensity map of A_g^2 peak ($\sim 120 \text{ cm}^{-1}$) with the polarizers set to horizontal (a) and vertical (b) position. **c-d**, Full Raman spectra in domain A and B under 0° and 90° polarization angles. **e-g**, Polarization-angle-dependent polar plots of the Raman B_{3g} , A_g^2 and A_g^3 peaks, taken from a spot in domain A (red curve) and another spot in domain B (blue curve).

5. Minor suggestions: the figures can be further polished to facilitate the reading. In Figure 2 and 4, the labels are quite small and the fonts are inconsistent; some panels (such as Figure 5a) are not very clear and the resolution or contrast can be improved.

Authors' response:

We thank the reviewer's suggestions on the aesthetics of the figures. We have adjusted the fonts and resolution accordingly.

Reviewer #3 (Remarks to the Author):

This work explores multiferroic phases of tin selenide (SnSe), where the inversion symmetry breaking is maintained in AA stacked multilayers over a range of thicknesses. This structure results in out of plane

ferroelectric behavior, in addition to the previously observed in-plane polarization. The key point is that the observed OOP ferroelectric response offers a path to active devices and the voltage dependence observed suggests a potential for low-voltage operation of those devices enabled.

Authors' response:

We thank the reviewer for recognizing the novelty and significance of our work.

This work would be greatly improved if the authors focus more directly on the OOP polarization phase induced and the materials development steps needed to result in these phases. It is not clear from the text how easily or how large the AA polarized films can be produced. This needs to be more clearly addressed. The formation of this homolog is key to device applications particularly low voltage operation.

Authors' response:

We appreciate the reviewer's constructive comments, mainly about the further elaboration of OOP polarization phase, and the formation of this phase. And here we address these issues as follows:

First, the OOP polarization phase is indeed the key observation in this work, and we apologize for the fact that our original manuscript might not highlight this finding enough. The realization of OOP polarized phase involves two key factors: AA stacking which preserves the inversion symmetry breaking of the monolayer, and the observed collective sliding which generates OOP polarization in addition to the original IP polarization. This is the major focus of our work as explained in the following logic. The first section of the main text (page 4, and Fig. 1b-c, e-f) is focused on verifying the non-centrosymmetric AA stacking based on optical SHG characterizations, since the broken inversion symmetry is a prerequisite for ferroelectric polarizations to form. Then the next section (page 5) is dedicated to the PFM characterization of both IP and OOP switching, with an emphasis on the capability of low-voltage OOP switching (Fig. 2c-f). The following section discusses the mechanism that generates the OOP behavior. Here, the detailed analysis on the interlayer sliding (page 6, Fig. 3, Fig. 4) is essential to our explanation because the sliding is actually the most critical factor that directly induces OOP polarization.

Following the reviewer's suggestion, we made the following changes in the main text to emphasize the details about OOP polarization phase and its intrinsic connection with the sliding.

*[Page 5, Line 32] Sub-title: **Sliding induced OOP polarization and its coupling mechanism***

[Page 6, Line 20] The direction of OOP polarization is determined by the sliding direction and locked to the IP polarization state, thus accounts for the observed OOP ferroelectricity. With a leftward sliding as in Fig. 4a, when the IP state is switched from $+P_{IP}$ to $-P_{IP}$, the OOP dipoles flip from $-P_{OOP}$ to $+P_{OOP}$

correspondingly, and vice versa. Such coupling arises as a direct consequence of AA stacking and sliding structure, as the $(-P_{\text{OOP}}, +P_{\text{IP}})$ and $(+P_{\text{OOP}}, -P_{\text{IP}})$ ground states are intrinsically associated with a C_{2y} symmetry operation in this fashion (Supplementary Fig. S10).

[Page 6, Line 26] To further understand the sliding induced OOP polarization and the coupled switching process, ...

[Page 6, Line 30] ..., and the system tends to relax into either of these energy-favorable states, agreeing well with the experimentally observed collective sliding structure. Following the ferroelectric switching process depicted in Fig. 4a with the lowest-energy amount of interlayer sliding, both the IP and OOP polarization values are calculated along the ferroelectric switching path using the nudged elastic band (NEB) method (Fig. 4c), which reveals a coupled switching behavior, where the IP polarization and OOP polarization changes synchronously.

Second, we discuss the materials development steps for growing the OOP polarized phase. We apologize that the details presented in the original manuscript were not clear enough. The growth of SnSe is performed in standard single-zone tube furnace with commercial SnSe powders as PVD source. The center temperature around the source region is typically 600-650 °C, which is higher than similar reports. The deposition temperature around the substrate is about 250 °C. The flow rate is kept around 60 sccm with a low pressure of 60 mTorr. We also note that certain parameters seem to favor the formation of thinner flakes. When growing thinner SnSe flakes, the source temperature is reduced to 550-600 °C, the deposition temperature is maintained around 250 °C, and the flow rate is raised to 100 sccm with the pressure increased to ~150 mTorr accordingly. The typical lateral size for 5-10 nm thick flakes is 5-10 μm , and 10-20 μm for relatively thicker samples.

Following the reviewer's advice, we have added all this detailed information into the manuscript:

[Page 5, Line 15] By increasing the carrier gas flow and decreasing both the growth temperature and growth time, we were able to suppress the layer-by-layer growth, and thus, to substantially decrease the thickness of SnSe sheets down to nanometer scale.

*[Page 19, Line 27] **Methods** section: ... The source temperature at the center of the tube furnace is set to 600-650 °C, and the evaporated SnSe is deposited downstream onto the substrate at a low temperature of 250 °C. To favor the formation of thinner flakes, the center temperature is reduced to 550-600 °C, with increased flow rate of 100 sccm and pressure of 150 mTorr, along with reduced growth time. The typical lateral size for 5-10 nm thick flakes is 5-10 μm , and 10-20 μm for relatively thicker flakes.*

We empirically conclude that reducing the temperature and increasing the gas flow tend to produce thinner flakes, and increasing the growth duration enlarges the flake size. We suspect that the temperature gradient around the substrate in the downstream of the tube furnace is one of the factors that produces the AA stacked phase. We note that the intrinsic growth mechanism for the formation of OOP polarized phase is still worth additional dedicated investigation in the future, but the materials development steps have now been provided.

The OOP behavior is the real "new" result here. It needs to be focused on in more detail than exists in the current manuscript. The formation of the AA state is key. It's not clear from the manuscript if AA is strain induced or naturally occurring. Do the author randomly find AA platelets created in their growth process? Are the AA stacks uniform and stable?

Authors' response:

We thank the reviewer for pointing out various concerns about AA stacking. We listed the following point-by-point evidence to demonstrate the uniformity of our samples.

(1) The AA stacked phase is naturally formed in the growth step, without external strain modulation. The optical SHG and the OOP ferroelectric measurements are performed on as-grown pristine samples without transfer or other type of fabrication, which confirms that the symmetry breaking and the OOP ferroelectricity are intrinsic. In addition, we believe the lattice mismatch induced strain between SnSe and the silicon substrate is not the dominant factor of AA stacking, because they are coupled with van der Waals interaction which is typically 1-2 orders of magnitude weaker than covalent bonds. The non-epitaxial van der Waals growth is thus less impacted by the lattice mismatch induced strain. In addition, although the sample-to-substrate coupling is weak, it still poses a certain amount of strain to the atomic layers that are directly in contact with the substrate. However, due to the interlayer van der Waals coupling in the SnSe flakes, such effects are released within just the bottom few layers, and the rest of the layers are not involved.

(2) The AA stacking proves to be uniform in the samples we tested. As shown in Fig. R10, we perform additional cross-section HRTEM to demonstrate the uniformity of AA phase. The AA stacking is verified in multiple samples from different batches. (Note that due to the variation in the FIB preparation process, and the slight misalignment of the zone axis during imaging, the image quality varies.) Apart from the direct HRTEM imaging, selected area electron diffraction (SAED) from the [010] orientation can also verify the AA stacking (Fig. 3b-c in the manuscript). So we take [010] diffraction patterns on two more samples as shown in Fig. R11, and they are in good consistency with both the simulation (Fig. R11a) and the result in main text (Fig. 3b). SAED is typically performed over a region of hundreds of nanometers to a few microns, which confirms the uniformity of AA stacking on a larger scale.

Figure R10 | Cross-sectional HRTEM images of the (010) crystal plane of SnSe. A schematic crystal model of AA-stacked SnSe is overlapped onto the image to show good correspondence.

Figure R11 | SAED patterns of SnSe along the [010] zone axis. *a*, Simulated result. *b-c*, Experimental results.

(3) The AA stacking is stable in our experiments, as the measured results weeks after growth are consistent with those directly after growth. Also, the cross-section TEM were performed on both pristine samples and samples after ferroelectric poling, and the stacking results are consistent. We would also like to clarify that the OOP polarized phase we observed is not a pure AA stacking, but with a collective interlayer sliding as well. From the total energy of the system, as shown in Fig. 4b, the OOP ferroelectric states correspond to the two local energy minima, suggesting that the collective interlayer sliding helps stabilize the AA phase. Here we also provide an additional set of sliding analysis in Fig. R12 (also added in *Supplementary Fig. S9*), proving that the collective sliding behavior which determines the OOP ferroelectric switching is global.

Figure R12 |TEM analysis of the interlayer sliding in an additional sample image. **a**, Analysis of the atomic positions from the side view imaged by HRTEM. Atoms are classified and marked by the same color scheme as in Fig. 3d. **b**, The horizontal position of atoms from different layers, confirming a collective interlayer sliding.

They show low, sub -0.3V ferroelectric switching, important for low-power consumption applications. In addition, this work also reports how stacking controls both IP and OOP ferroelectric properties and the ability to couple these polar interactions.

Authors' response:

We appreciate the reviewer's positive evaluation on these discoveries.

This manuscript needs revision to focus more directly on the viability of creating AA stacked materials for device applications, again the OOP performance and the low voltage operation are the novel findings in this work. This refocusing is needed before this work adequate for publication.

Authors' response:

We are thankful for the reviewer's suggestions on improving our manuscript. As discussed above, we have systematically revised the manuscript and addressed all specific comments, including the detailed material preparation steps, the uniformity and yield of the OOP polarized phase, and more emphasis on the OOP performance with elaborated explanation of its switching mechanism. Meanwhile, the collective sliding behavior is considered a fundamental mechanism that generates the OOP polarization and helps stabilize the AA stacked phase we are focused on. It also introduces a novel IP and OOP coupling which might create more functionalities for device applications. We believe such discussion is essential for understanding the

OOP polarized phase and fits into the scope of the OOP performance. This work is conceived as a material-property-oriented project, where we'd like to report the novel behaviors that we observe in SnSe, while the intrinsic growth mechanism or the specific device application scenarios deserve separate exploration by future efforts. We hope we can convince the reviewer that this revised manuscript meets the publishing criteria of *Nature Communications*.

References

1. Liu, F. et al. Room-temperature ferroelectricity in CuInP2S6 ultrathin flakes. *Nature Communications* 7, 12357 (2016).
2. Io, W. F. et al. Direct observation of intrinsic room-temperature ferroelectricity in 2D layered CuCrP2S6. *Nat Commun* 14, 7304 (2023).
3. Zhou, Y. et al. Out-of-Plane Piezoelectricity and Ferroelectricity in Layered α -In₂Se₃ Nanoflakes. *Nano Lett.* 17, 5508–5513 (2017).
4. Fei, Z. et al. Ferroelectric switching of a two-dimensional metal. *Nature* 560, 336–339 (2018).
5. Jiang, Y. et al. Enabling ultra-low-voltage switching in BaTiO₃. *Nat. Mater.* 21, 779–785 (2022).
6. Wang, Y. et al. A stable rhombohedral phase in ferroelectric Hf(Zr)_{1-x}O₂ capacitor with ultralow coercive field. *Science* 381, 558–563 (2023).
7. Yang, Q. et al. Ferroelectricity in layered bismuth oxide down to 1 nanometer. *Science* 379, 1218–1224 (2023).
8. Zhang, X. & Peng, B. The twisted two-dimensional ferroelectrics. *Journal of Semiconductors* 44, 011002 (2023).
9. Springolo, M., Royo, M. & Stengel, M. Direct and Converse Flexoelectricity in Two-Dimensional Materials. *Phys. Rev. Lett.* 127, 216801 (2021).
10. Kumar, S., Codony, D., Arias, I. & Suryanarayana, P. Flexoelectricity in atomic monolayers from first principles. *Nanoscale* 13, 1600–1607 (2021).

11. Zheng, J.-D. et al. Flexoelectric effect induced p–n homojunction in monolayer GeSe. *2D Mater.* 9, 035005 (2022).

Reviewer #1 (Remarks to the Author):

The author answered my questions and supplemented relevant data.

Authors' response: We are very thankful for the reviewer's positive evaluations of our revision.

Reviewer #2 (Remarks to the Author):

In the revised version, the authors have provided sufficient experimental characterizations and theoretical analyses to address the comments raised by the reviewer. The questions are well explained, and the manuscript has been improved accordingly. I thus recommend its acceptance to Nature Communications.

Before publication, the authors may check the following minor suggestions:

1. The four coupled polarization states and the associated symmetry operations are thoroughly discussed in the responses (2nd question). The authors should include some of these descriptions into the SI as well.
2. The format of the images should be kept as consistent as possible, like the scale bar in TEM images, should be preserved as uniformly as possible.

Authors' response: We appreciate the reviewer's recommendation of our manuscript. We have added a supplementary note in SI and adjusted the figures accordingly.

Reviewer #3 (Remarks to the Author):

I believe the authors have addressed my primary concerns. Details on creation of the OOP polarization phase have been included. The authors state that they have included uniformity and yield information on the OOP phase in the response, but I failed to identify that language in the main manuscript. It's still not clear to me if they can efficiently synthesize that phase for device applications.

The work is of significance to the field. I believe it should be accepted for publication.

Authors' response: We appreciate the reviewer's recommendation of our manuscript. We have included the discussion of uniformity and yield in the manuscript following the reviewer's suggestion.